# A thousand-genome panel retraces the global spread and adaptation of a major fungal crop pathogen

Alice Feurtey[1,2,3], Cécile Lorrain [2], Megan C. McDonald[4,5], Andrew Milgate [6], Peter S. Solomon [4], Rachael Warren [7], Guido Puccetti[1,8], Gabriel Scalliet [8], Stefano F. F. Torriani[8], Lilian Gout[9], Thierry C. Marcel [9], Frédéric Suffert[9], Julien Alassimone [2], Anna Lipzen[10], Yuko Yoshinaga[10], Christopher Daum [10], Kerrie Barry[10], Igor V. Grigoriev [10,11], Stephen B. Goodwin [12], Anne Genissel [9], Michael F. Seidl[13,14], Eva H. Stukenbrock [3,15], Marc-Henri Lebrun[9], Gert H. J. Kema [13], Bruce A. McDonald [2] & Daniel Croll [1] ✉

Human activity impacts the evolutionary trajectories of many species worldwide. Global trade of agricultural goods contributes to the dispersal of pathogens reshaping their genetic makeup and providing opportunities for virulence gains. Understanding how pathogens surmount control strategies and cope with new climates is crucial to predicting the future impact of crop pathogens. Here, we address this by assembling a global thousand-genome panel of *Zymoseptoria tritici*, a major fungal pathogen of wheat reported in all production areas worldwide. We identify the global invasion routes and ongoing genetic exchange of the pathogen among wheat-growing regions. We find that the global expansion was accompanied by increased activity of transposable elements and weakened genomic defenses. Finally, we find significant standing variation for adaptation to new climates encountered during the global spread. Our work shows how large population genomic panels enable deep insights into the evolutionary trajectory of a major crop pathogen.

Human activity has broken down natural barriers to gene flow for many species through trade and travel. Reshaped species distributions helped spread invasive plants and pathogens[1]. A major contributor to the range expansion of pathogens is the distribution of suitable host species. Pathogens and their hosts often share a common evolutionary history either through co-evolution or through shared constraints of their common environment[2,3]. Discrepancies in the evolutionary history of hosts and their pathogens can be caused by host jumps,

[1]Laboratory of Evolutionary Genetics, Institute of Biology, University of Neuchâtel, CH-2000 Neuchâtel, Switzerland. [2]Plant Pathology, D-USYS, ETH Zurich, CH-8092 Zurich, Switzerland. [3]Max Planck Institute for Evolutionary Biology, Plön, Germany. [4]Division of Plant Science, Research School of Biology, The Australian National University, Canberra, ACT, Australia. [5]School of Biosciences, Institute of Microbiology and Infection, University of Birmingham, Birmingham, UK. [6]NSW Department of Primary Industries, Wagga Wagga Agricultural Institute, Pine Gully Road, Wagga Wagga, NSW 2650, Australia. [7]The New Zealand Institute for Plant and Food Research Limited, Lincoln, New Zealand. [8]Syngenta Crop Protection AG, CH-4332 Stein, Switzerland. [9]Université Paris Saclay, INRAE, UR BIOGER, 91120 Palaiseau, France. [10]US Department of Energy Joint Genome Institute, Lawrence Berkeley National Laboratory, Berkeley, CA 94720, USA. [11]Department of Plant and Microbial Biology, University of California Berkeley, Berkeley, CA 9472, USA. [12]USDA-Agricultural Research Service, West Lafayette, IN, USA. [13]Wageningen University and Research, Laboratory of Phytopathology, Wageningen, The Netherlands. [14]Utrecht University, Theoretical Biology and Bioinformatics, Utrecht, The Netherlands. [15]Environmental Genomics, Christian-Albrechts University of Kiel, Kiel, Germany. ✉e-mail: daniel.croll@unine.ch

significant differences in gene flow, or local adaptation. Agricultural pathogens have often emerged during the domestication of their host species[4] causing significant threats to food production. Increased global trade of agricultural products has precipitated serious disease outbreaks over the past decades[5]. Crop pathogens are exposed to globally homogeneous host conditions created by planting genetically similar crop cultivars and application of similar pesticidal compounds to control diseases[6,7]. Furthermore, climate change reshapes the geographic distribution of pathogen species, with poleward range expansions being suspected since the 1960s[8]. Range expansions may lead to significant changes in the genetic make-up of pathogen species by founder effects and shifting barriers to gene flow[9]. Understanding how emerging pathogens surmount control strategies and cope with climate adaptation is crucial to predict the future impact of crop pathogens in a changing world.

Outbreaks of fungal diseases on crops are reported regularly across continents[5]. In addition to episodic damage, most crop pathogens are endemic and continuously reduce yields. The ascomycete *Zymoseptoria tritici* is a major pathogen of bread and durum wheat, causing the disease Septoria tritici blotch, which is now reported in most wheat-growing regions and causes significant damage[10]. The center of origin of *Z. tritici* is located in the Middle East, where sister species are found to infect wild grasses[11]. The emergence of *Z. tritici* was concomitant with the domestication of wheat[11]. The timing and shared geographic origin of the pathogen and domesticated wheat strongly suggests coevolution between the two species. The pathogen harbors extensive standing variation from individual infected leaves to large agricultural regions[12,13]. As a consequence, the pathogen showed rapid responses across all major wheat-producing areas to overcome host resistance and gain tolerance to fungicides in less than a decade[10]. Population genomic analyses showed that rapid adaptation of the pathogen was facilitated by parallel evolution across geographic regions[14,15]. However, a comprehensive picture of pathogen dispersal and adaptation across the global distribution range is lacking.

Here, we assembled over one thousand genomes of the fungal crop pathogen *Z. tritici* to retrace worldwide invasion routes out of its Middle Eastern origin and identify ongoing genetic exchange among major wheat-producing regions. We show that the global expansion was accompanied by increased activity of transposable elements and weakened genomic defenses. Finally, we identify standing genetic variation for adaptation to new climates encountered during the global spread.

## Results

### Global genetic structure of the pathogen tracks the historical spread of wheat

We assessed the evolutionary trajectory of the pathogen in conjunction with the history of global wheat cultivation (Fig. 1a). For this, we assembled a worldwide collection of *Z. tritici* isolates from naturally infected fields (Fig. 1b). We selected isolates covering most wheat production areas, both in the center of origin of the crop (i.e., the Fertile Crescent in the Middle East), and in areas where wheat was introduced during the last millennia (i.e., Europe and North Africa), or last centuries (i.e., the Americas and Oceania; Fig. 1c). We called variants in a set of 1109 high-quality short-read resequencing datasets (Supplementary Data 1, 2) covering 42 countries and a broad range of climates. Using a joint genotyping approach, we produced raw variant calls mapped to the telomere-to-telomere assembled reference genome IPO323. To assess genotyping accuracy, we used eight isolates with replicate sequencing data to analyze discrepancies. We adjusted quality thresholds targeting specifically the type of genotyping errors observed in our data set (Fig. S1). The improved filtering yielded 8,406,818 high-confidence short variants (short indels and SNPs). The final variant set included 5,578,488 biallelic SNPs corresponding to 14.1% of the genome.

We tested whether global diversity patterns of pathogen populations are likely a consequence of the history of wheat cultivation. We first performed unsupervised clustering of genotypes and identified eleven well-supported clusters (Fig. 2a, Figs. S2,3). Over 90% of the genotypes were clearly assigned to a single cluster (Fig. 2a, Supplementary Data 3). Two clusters were identified among genotypes originating from the pathogen center of origin, distinguishing collections from Iran and Middle Eastern regions. Genotypes from Africa and Europe split into two distinct genetic clusters without any apparent secondary structure within clusters. This lack of any fine-scale structure is remarkable given the extensive geographic sampling of European genotypes and suggests extensive gene flow within the continent. Genotypes from Oceania grouped into three distinct clusters marked by collections from Tasmania, the Australian mainland, and New Zealand. Genotypes from North America formed two clusters along a North-South separation. Finally, South American genotypes formed two clusters split along the Andes (Chile versus Argentina and Uruguay). Some uncertainty exists in the assessment of regional population structure by low coverage of major wheat-producing countries such as Russia and Ukraine. Septoria tritici blotch is only sporadically reported in China. In complementary analyses, we found that a phylogenetic network accounting for the high frequency of recombination consistently reflected the global population structure (Fig. S4). A principal component analysis of all genotypes confirmed the nested genetic structure with differentiation at the continent level, subdivisions within some continents and the existence of admixed genotypes (Fig. 2b, Fig. S5).

We analyzed the history of population splits and admixture using allele frequency information (Fig. 2c). The analyses largely supported a genetic structure shaped by the introduction of wheat across continents. The historical relationships between clusters show an early divergence of the Middle Eastern and North African clusters matching the early introduction of agriculture in these regions. Populations in Europe and the Americas share a similar time point of divergence consistent with extensive contributions of European genotypes to the Western hemisphere. Oceanian groups have diverged as a single branch from genotypes most closely related to extant European populations. Matching the introduction of wheat to Oceania from the European continent, the Australian and New Zealand pathogen populations share a common origin rooted in European genetic diversity. Populations from Australia show also a striking loss of diversity and higher linkage disequilibrium compared to European diversity consistent with a significant founder effect (Fig. 2d, e). Similarly, populations in South and North America have reduced genetic diversity compared to extant European populations as suggested previously based on Sanger sequencing[16]. The highest diversity was found in populations from Africa and the Middle East closest to the center of origin. Overall, the global genetic structure of the pathogen reveals multiple founder events associated with the introduction of wheat to new continents.

Ongoing gene flow among regions should lead to admixed genotypes. We found that nearly 10% of all analyzed genotypes showed contributions from at least two clusters. The most significant recent gene flow was detected between Middle Eastern/North African clusters and European clusters in North Africa (i.e., Algeria and Tunisia) as well as Southern and Eastern Europe (i.e., France, Italy, Hungary, Ukraine, Portugal, and Spain; Supplementary Data 3). We found a particularly high incidence of recent immigration in a durum wheat population in the south of France. The population consisted only of hybrids or atypical genotypes suggesting either recent migration from North Africa or host specialization on durum wheat varieties. Additionally, we found hybrid genotypes with European ancestry in both North America and in Oceania. The relatively balanced ancestry proportions in these hybrids suggest very recent gene flow dating back to only a few generations. We further investigated past gene flow between clusters

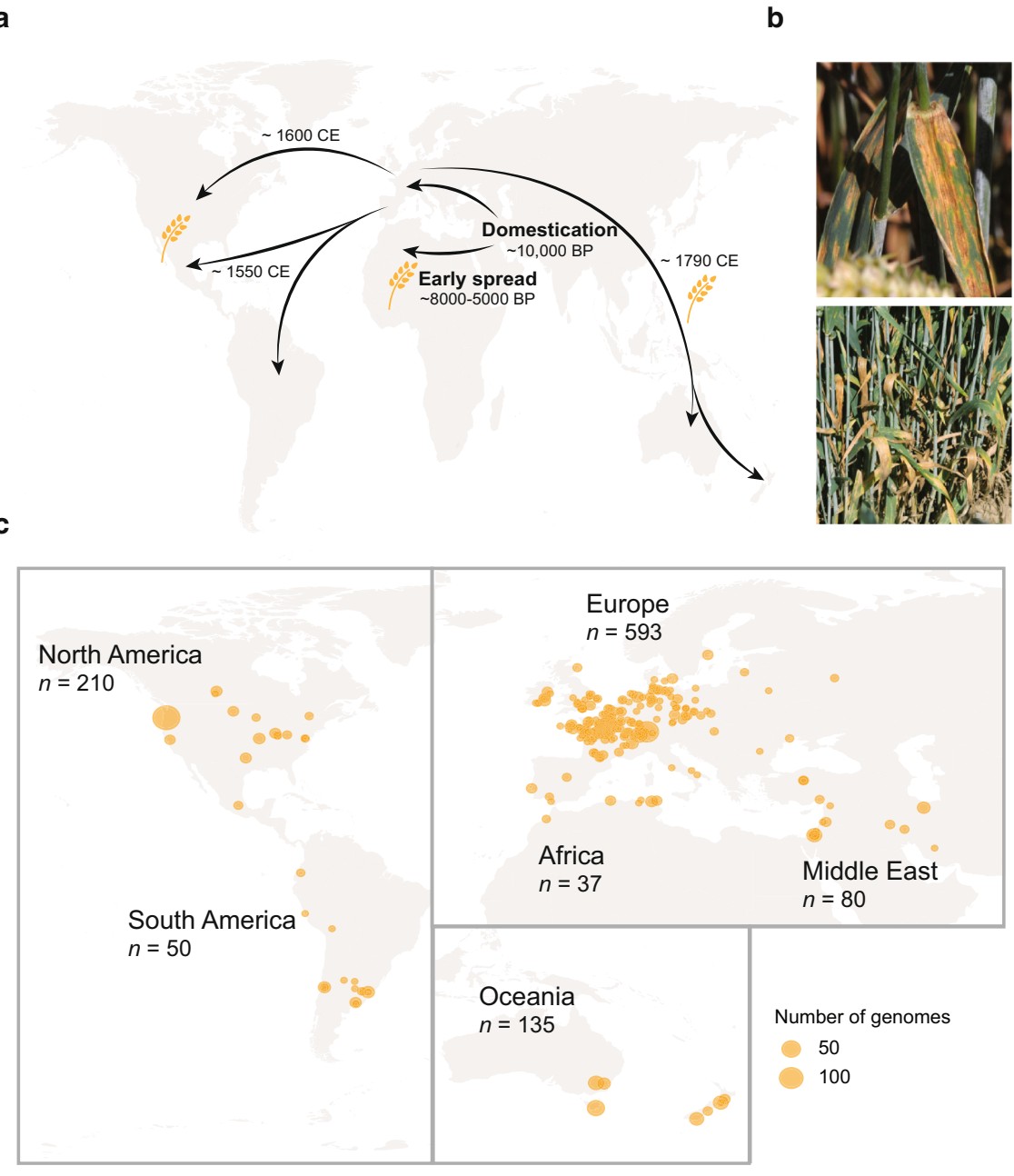

**Fig. 1 | Global sampling of the wheat pathogen *Zymoseptoria tritici* retracing the historical spread of its host. a** Schematic representation of the introduction of wheat across continents. **b** Septoria tritici blotch symptoms caused by *Z. tritici* on wheat leaves. Pictures taken by B. A. McDonald, ETH Zurich. **c** Map of the sampling scheme for the global collection of 1109 isolates for whole-genome sequencing.

by allowing Treemix to infer migration events, thus creating a population network (Fig. S6a–d). Three distinct recent migration events were best explaining the data with specific migration routes from the Middle East/African clusters to North America, from an Australian cluster to South America and between two Oceanian clusters (Fig. S6d). However, the migration events did not affect the overall shape of the inferred population tree (Fig. 2c, Fig. S6b–d). To better understand effects of long-distance gene flow, we investigated the relationship between relatedness among genotypes (i.e., identity-by-state) and geographic distance. At the continent level, we observed a negative relationship between identity-by-state and geographic distance (Fig. S7). The wide distribution of identity-by-state values shows that although closely related isolates tend to be found at closer geographic distance, distantly related isolates can be found at both far and close

geographic distances. Long-distance migration events are most likely caused by international trade similar as for other crop pathogens[17–19]. In combination, our findings show an important role of long-distance dispersal impacting the genetic make-up of populations from individual fields to continental scale genetic diversity.

**Relaxation of genomic defenses against transposable elements concurrent with global spread**

Transposable elements (TEs) are drivers of genome evolution. In *Z. tritici*, TE activity created beneficial mutations for fungicide resistance and virulence on the wheat host[20,21]. Rapid recent adaptation of the pathogen has benefitted from the activity of TEs with consequences for genome size[22]. Unchecked transposition of TEs can be deleterious and an array of defenses mechanisms has evolved to counteract their

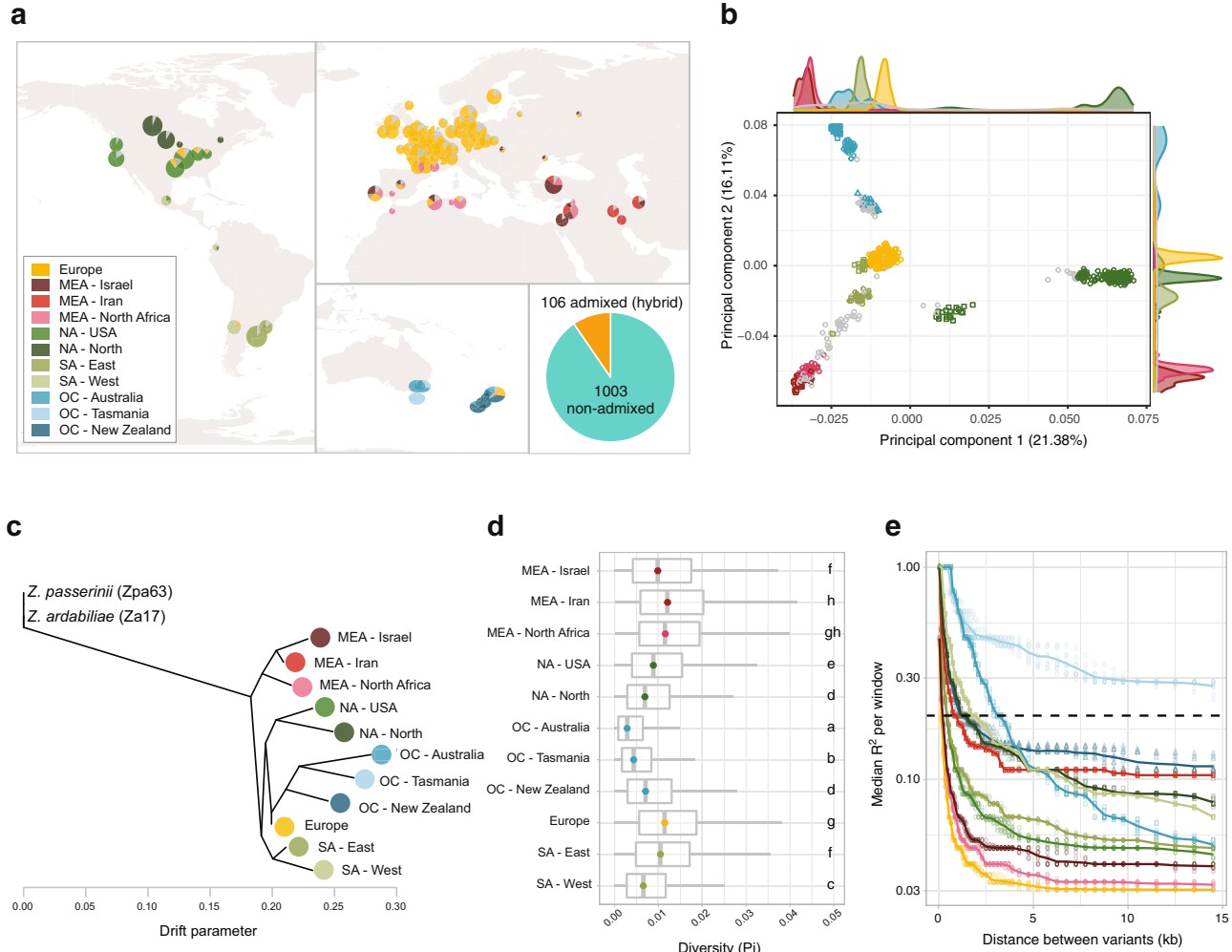

**Fig. 2 | Global genetic structure based on 1109 genomes. a** Map of the genetic clustering based on a thinned genome-wide SNP dataset using sNMF. Each color represents a different genetic cluster, and the sizes of the slices represent the average attribution to the cluster across the isolates from each location. Fractions representing less than 10% of all genotypes of a location were colored in grey to improve clarity. The large pie chart outside of the map represents the proportion of isolates assigned clearly (≥75%) to a single genetic cluster (pure; in teal) and isolates identified as hybrids (admixed) between clusters (in yellow). Names of the clusters include an abbreviation of continents and a more precise geographical location (MEA: Middle East and Africa; NA: North America; SA: South America; OC: Oceania). **b** Principal component analysis, showing the first and second component (PCs) based on a subset of variants. Colors and shapes indicate the genomic clusters identified with the sNMF method (with hybrids in grey). The marginal distributions represent the distribution for each PC. PCs 1 to 8 are shown in Fig S4. **c** Population tree based on Treemix, rooted using two genomes from the sister species *Z. passerinii* and *Z. ardabiliae*. The colors are the same as in the previous panels and only samples which were fully assigned to a cluster were used. **d** Diversity estimated with using *pi* per genetic cluster. The boxplots are ordered according to the tree of panel. **c.** The lower and upper hinges correspond to the first and third quartiles, the whiskers to the largest value are within 1.5 times the inter-quartile range, and the central horizontal line defines the median. **e** Linkage disequilibrium ($r^2$) between variants per genetic cluster. Colors are identical among panels.

activity both at the genomic and epigenetic level including targeted mutations and silencing[23]. To analyze the effectiveness of genomic defenses against active TEs, we screened all genomes for evidence of TE insertions. We mapped short-read sequencing data on the reference genome and a species-specific TE sequence library. We classified evidence for TEs in each of the analyzed isolates as reference TEs (i.e. also present in the reference genome) and non-reference TE (i.e. absent). Detected TEs among isolates were binned into loci (width 100 bp) to account for uncertainties about the precise mapping of the insertion point. We found that the frequency spectrum of TE insertions is heavily skewed towards low frequencies with 77% of TE insertions being found in single isolates (~0.1% frequency) and 96% of insertions were found in ten or fewer isolates (<1%; Fig. S8a) consistent with strong purifying selection. The *Z. tritici* genome contains both core and accessory chromosomes (i.e. chromosomes not shared among all isolates of the species; Fig. 3a)[24]. Accessory chromosomes have higher TE densities (15–33% vs. 5.5–19%[24]; Fig. 3b) reflecting lower selection pressure on accessory chromosomes[25]. Beyond this, accessory chromosomal regions are broadly differentiated from core regions based on sequence, transcription and epigenetic feature sets (Fig. 3c). The primary differentiator (i.e., the first principal component) separated euchromatic and heterochromatic (H3K9me2) chromosomal segments. These differences in epigenetic marks were matching TE density variation, consequences of genomic defenses (i.e., repeat-induced point mutations), and GC content. The number of TE insertions was not correlated with GC content but was positively correlated with the presence of facultative heterochromatin (H3K27me3) and negatively correlated with euchromatin marks. H3K27me3 is also a hallmark of accessory chromosome segments consistent with previous findings[26–28]. Short insertion/deletion (indel) polymorphism was

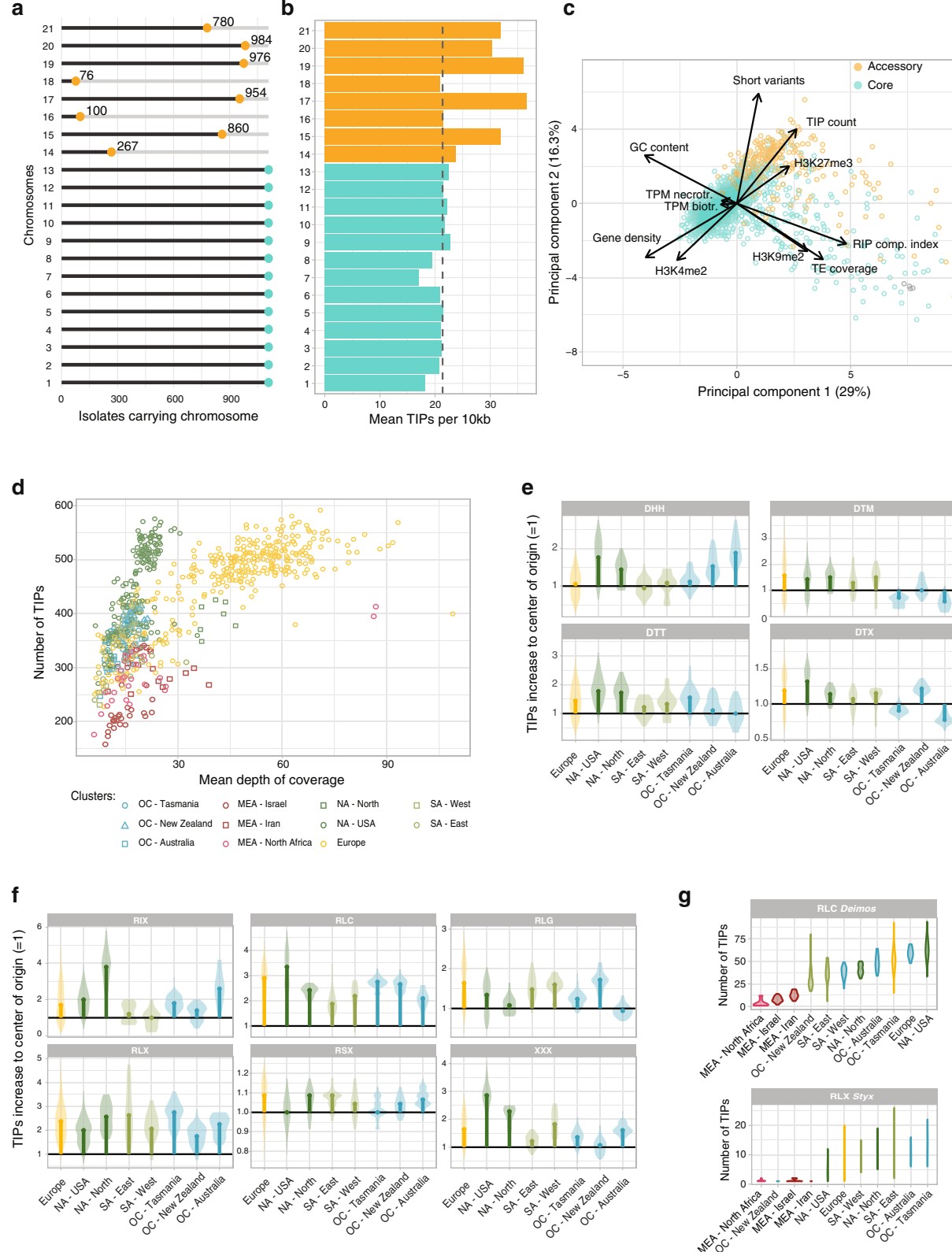

positively correlated with the heterochromatic accessory regions, consistent with purifying selection acting against indels mostly in gene-dense regions.

The pathogen genome shows variation in TE content among populations from different continents, suggesting that the TE content was influenced by the colonization history of the species[22,29]. Hence, we tested whether recently established populations were differentiated

from center-of-origin populations. We found that TE insertion polymorphisms (TIPs) are most often specific to a genetic cluster with only 31% of insertions shared by two or more populations (Fig. S8b). We used TIPs as a genetic marker and found that population differentiation was matching differentiation assessed using short variants (Fig. S8c). Hence, the population history of the pathogen was an important factor shaping TE content. We also found that the per-individual TE

**Fig. 3 | Transposable element expansion outside of the pathogen's center of origin. a** Presence of each chromosome in the 1109 isolates assessed by depth of coverage. Yellow colors in panels **a–c** represent the known accessory chromosomes while the core chromosomes are shown in teal. **b** Mean number of transposable element insertion polymorphisms (TIPs) per 10-kb window for each chromosome. The dotted line represents the genome-wide mean. **c** Principal component analysis using genomic, epigenomic and transcriptomic information per 10-kb window as well as number of variants (short variants and TIPs). **d** Dot plot showing the number of TIPs and the genome-wide depth of coverage for each sequenced isolate. The color and shape differentiate the genetic clusters identified based on the genome-wide short variants. Names of the clusters include an abbreviation of continents and a more precise geographical location (MEA: Middle East and Africa; NA: North America; SA: South America; OC: Oceania). **e** Violin plots of the number of TIPs per DNA transposon (class II) TE family and per cluster. TIP numbers were normalized by the median from the three clusters found in the center of origin. Only TE families with a median number of insertions above eight are represented. **f** Violin plots of the number of TIPs per retrotransposon (class I) TE family and per isolate with identical normalization as in panel **e**. **g** Number of TIPs per isolate for two TEs showing an increase out of the center of origin.

content has increased in most areas outside of the center of origin. TE detection using short read sequencing data is constrained by sequencing depth, hence we accounted for depth in our comparative analyses of TE content (Fig. 3d). The TE content of the Middle Eastern and African genomes was lower than in any other regions ranging from 164–394 (median = 302) and 184–429 TIPs (median = 300), respectively, including only samples with coverage below 35. In contrast, the Oceanian, North American and South American genomes contained among the highest TE numbers (258–513, median = 373; 283–513, median = 462; 237–476, median = 375 TIPs, respectively) and European genomes showing intermediate TE counts (209–669, median = 334 TIPs, including only samples with a coverage below 35).

The increase in TE content was broadly spread among superfamilies and included both DNA transposons and retrotransposons (Fig. S9, 10). Some superfamilies showed particularly strong signatures of expansion outside of the Middle East and Africa including *Copia* retrotransposons (RLC) and other LTR retrotransposons (RLX) (Fig. 3e, f). The RLC expansion is mainly driven by the *Deimos* element with an increase from 11.3 copies on average in genomes from the Middle East and Africa to an average of 50.8 copies per genome outside the center of origin (Fig. 3g and S9, 10). The RLX superfamily expansion is driven by the Styx element, generally more present outside the center of origin (Fig. 3g and S9,10). The *Styx* activity was shown to affect asexual reproduction on the host[30]. We also observed population-specific expansions such as the LINE (RIX) *Juno* element mostly present in North American and Australian populations, the Helitron (DHH) *Fatima* expanded primarily in the Australian population, and the DTX miniature inverted-repeat TE (MITE) *Unicorn* most widespread in North American and New Zealand populations (Fig. S9,10). The broad increase in TE content outside of the center of origin suggests a relaxation of genomic defenses over the evolutionary history of the pathogen on wheat.

Many ascomycetes share a genome defense mechanism against TEs that can rapidly introduce targeted mutations into newly duplicated sequences, called repeat-induced point mutation (RIP)[31,32]. RIP machinery is active in genomes of *Z. tritici*, with high levels of RIP-like mutations identified in genomes from the center of origin and wild-grass-infecting sister species[29,33]. We analyzed the global panel for evidence of RIP-like mutations by reporting the RIP composite index. The median index is above 1 in TE sequences across all genetic clusters. However, we found that RIP strength varies considerably among genetic clusters with the strongest signatures found in genomes from Middle Eastern and African isolates (Fig. 4a). The Middle Eastern and African clusters tend to include genomes with both low TE content and strong RIP signatures (Fig. 4b, S8a, b). In more recently colonized regions, genomes showed a negative correlation between the strength of RIP signatures and the amount of TEs per genome ($p$ value $<2.2e-16$; slope of $6.7 \times 10^{-4}$ in all populations outside of the Middle-East and slope of $4.6 \times 10^{-4}$ in the European population only; Fig. 4b & S11a, b). The association between TE content and strength of RIP signature is consistent with genomic defenses being less capable to prevent new TE insertions following migration out of the center of origin.

How genomic defenses against TEs are modulated in fungi remains largely unknown. The RIP machinery is activated during sexual recombination in *Neurospora*[34], thus suggesting that reduced sexual reproduction could lower the efficacy of RIP. In *Z. tritici*, sexual reproduction can occur during the wheat growing season but most reproduction is thought to occur at the end of the season[35]. The rate of sexual reproduction is high in all populations as we found that the ratio of mating types was consistently close to 50:50 (Fig. S11c). Alternatively, the RIP machinery could have lost function in some populations. In this case, TEs should show bimodal signatures of RIP with old TEs carrying signatures of historical RIP activity and recent insertions with no or weak RIP signatures. Indeed, a high percentage of TEs shows only weak evidence for RIP (composite index <0.5) in genetic clusters outside of the Middle East and Africa (mean = 20.1%, range from 11.3–34.6%; Fig. S11d). By contrast, in the center of origin all genomes had less than 10% of TEs showing weak evidence for RIP (mean = 7.48%, min = 5.18%). We confirmed the bimodal RIP signatures by analyzing individual TEs in 20 chromosome-level assembled genomes (Fig. 4c). All genomes shared a major RIP composite index peak of ~2. A secondary peak indicative of TEs without RIP was found in genomes from the Americas, Oceania, and Europe. Despite the higher TE activity, we found no evidence that the loss in RIP functionality also led to high gene duplication events. The strength of RIP signatures is associated with the size of the TEs as small elements show little to no RIP signatures (Fig. 4d) consistent with the RIP machinery being inactive on small repeats[23]. In *Neurospora crassa*, two pathways mediate RIP with one dependent on RID and one on Dim2[34]. In *Z. tritici*, the Dim2 methyltransferase was functionally linked to the occurrence of RIP-like mutations in repeats including during mitosis[33,36]. Furthermore, the presence of a functional *dim2* copy is strongly correlated with lower GC content of TEs, hence deamination of cytosines[33,36]. The *dim2* gene was duplicated multiple times in some genomes causing loss-of-function mutations triggered by the RIP mechanism itself[33,37]. We found that the ancestral copy of *dim2* shared higher identity with the functional copy of *dim2* in the genomes of the Middle Eastern isolates than other populations (Fig. S11e). The European populations had both the largest range in the number of detected paralogs and the highest copy numbers overall (Fig. S11f). This is consistent with a deleterious runaway gene-duplication process affecting a molecular component of the RIP machinery, explaining the loss of RIP efficacy within the species.

## Adaptation to fungicides and changing climates along continental gradients

Globally distributed pathogens experience significant environmental heterogeneity that potentially constrains or facilitates future range expansions and adaptation. The use of pesticides across the globe to combat agricultural pathogens has triggered the parallel emergence of resistance with significant economic consequences[6]. To retrace the global emergence of fungicide resistance in *Z. tritici*, we analyzed mutations in resistance genes using isolates collected over three decades (1986 to 2016). This time span covers the introduction of several major fungicides to agricultural fields. Resistance to the ubiquitously used azole fungicides is often mediated by mutations in the *CYP51* gene[38]. Recent North American populations gained the Y137F mutation but not the I381V or V136A mutations rising in frequency in Europe

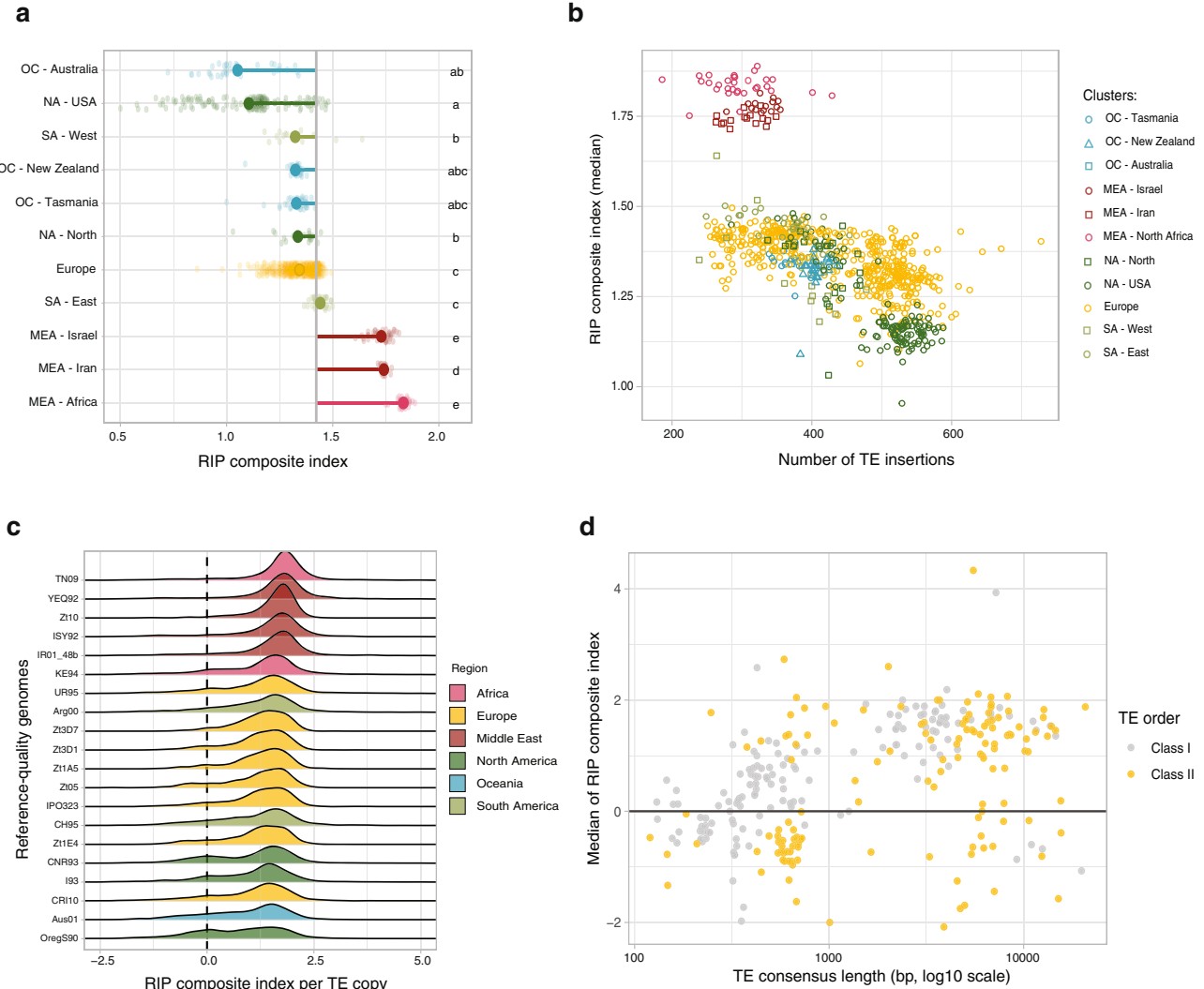

**Fig. 4 | Relaxation of genomic defenses against transposable elements. a** RIP composite index in the TEs for each isolate (small transparent dot) and as an average per genetic cluster (large opaque dot). Names of the clusters include an abbreviation of continents and a more precise geographical location (MEA: Middle East and Africa; NA: North America; SA: South America; OC: Oceania). The grey vertical line represents the overall average. **b** Dot plot representing the number of TIPs per genome compared to the median of the RIP composite in reads mapping on TE consensus sequences. The colors/shapes are the same as in panel **d**. **c** Distribution of RIP composite index per TE copy in 20 high-quality genome assemblies for the species. The dotted line at value 0 corresponds to no detected RIP signal. **d** Median of the RIP composite index of the reads aligned to a specific TE consensus as a function of the TE consensus length.

over the same period (Fig. 5a). European populations harbored the most diverse set of azole resistance mutations, consistent with the early and intense applications of this fungicide class. The I381V and V136A mutations occurred at high frequency since the early 2000s, whereas the S524T mutation was only observed at a low frequency with a delayed onset. Resistance arose later in Oceania and North America, consistent with the later application of azoles in those locations. No resistance mutations were detected in the Middle Eastern or African populations, matching the absence of azole treatments in those regions. We found similar geographic patterns for the E198AK mutation in the beta-tubulin gene associated with benzimidazole resistance as well as for the G143A mutation in the mitochondrial gene *cytb*, known to cause resistance to Quinone outside inhibitors fungicides (Fig. 5b). As expected from their more recent introduction, mutations related to resistance to succinate dehydrogenase inhibitors (SDHI) were only observed in the most recently sampled populations in Europe (Fig. S12). Overall, the global analyses of fungicide resistance signatures show how European populations consistently developed the first known mutations to newly introduced fungicide.

Changes in climatic conditions create complex challenges for plant breeders to create resilient crops[39]. Concurrently, pathogen populations are exposed to changes in temperature and humidity patterns. The historic spread of *Z. tritici* has likely created significant selection pressure to adapt to climates associated with the global range of wheat cultivation. Here, we analyze the genetic architecture of climate adaptation by mapping standing variation along climatic clines. The pathogen is endemic to regions with distinct climates, from the dry and warm conditions in the Middle East to the temperate oceanic climate of Western Europe and the humid continental climates of some North American locations. We performed genotype-environment association (GEA) mapping based on the climatic conditions of the sampling locations. We analyzed a total of 19 bioclimatic variables covering annual trends, seasonality, and extreme environmental factors, such as the maximum temperature of the warmest month or the precipitation of the driest quarter year (Fig. 5c, d, Supplementary Data 4).

We identified 1956 variants significantly associated with at least one climatic variable and 640 variants with a minor allele frequency

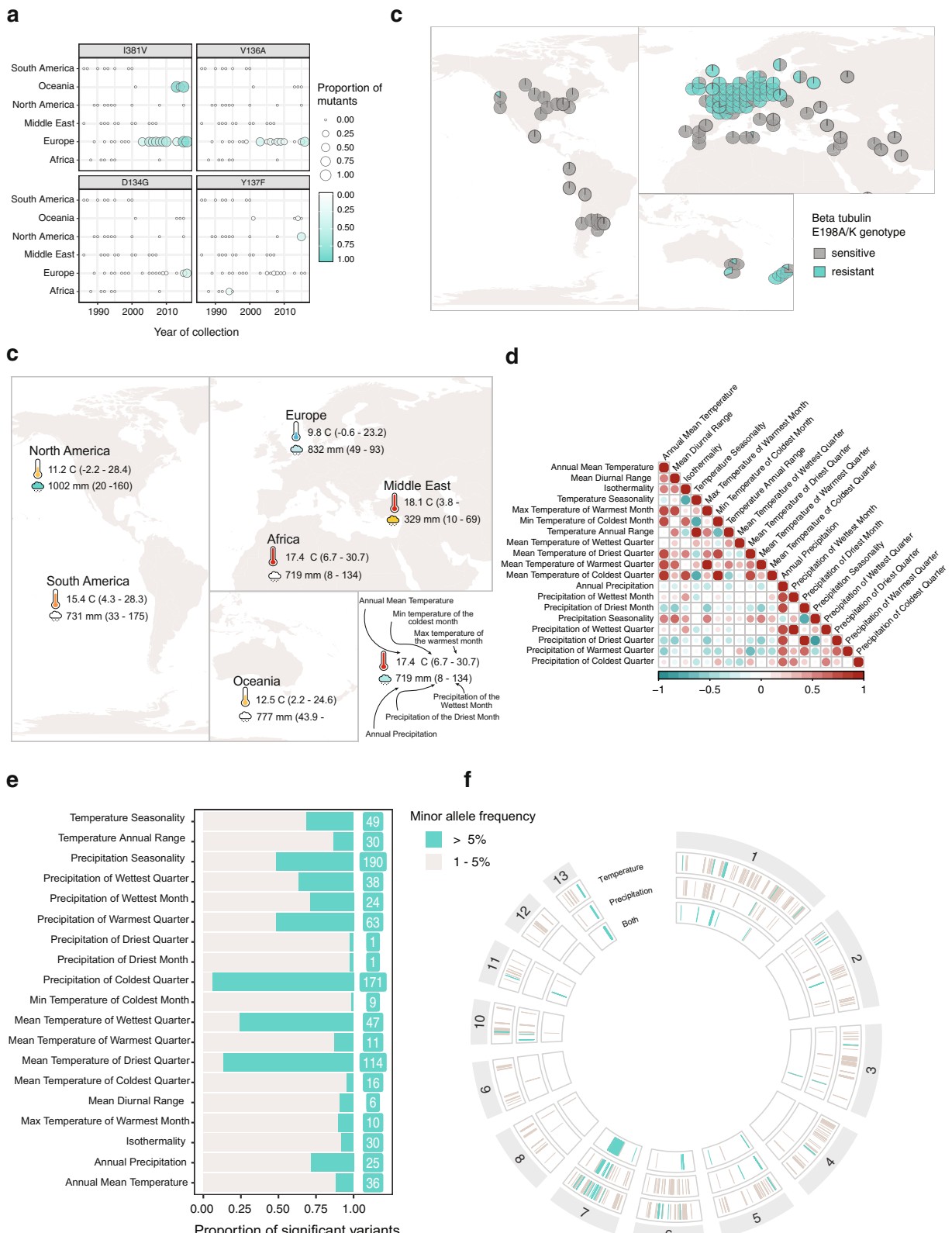

Fig. 5 | **Adaptation to fungicides and climatic gradients. a** Frequency of mutations underlying azole fungicide resistance across time and regions. **b** Map representing the allele frequency of known beta-tubulin resistance mutations (E198A/K) against benzimidazole fungicides. **c** Examples of bioclimatic variables used for the genotype-environment association analyses among regions. The values represent the average for each continent. **d** Correlation plot between the 19 analyzed bioclimatic variables. **e** Proportion and number of variants significantly associated with each bioclimatic variable and with a minor allele frequency (MAF) higher than 5%. **f** Genomic location of variants significantly associated with the bioclimatic variables grouped into three main categories (variables describing temperature, precipitation levels, and variables describing combined measures of temperature and precipitation).

(MAF) higher than 5%. The number of associated variants per climatic variable ranged between 36 and 541 (for BIO9 and BIO6 respectively), including 1–190 significant SNPs with a MAF >= 5% per climatic variable (Fig. 5e). We investigated whether variant classes were enriched in the set of significantly associated SNPs. Using permutations, we found that both non-synonymous and intergenic variants were more frequently associated with climatic variables than expected randomly while synonymous variants were significantly depleted (Fig. S13b). We found 187 genes that were in proximity or directly affected by variants associated with bioclimatic variables and with a MAF >= 5%, including 65 containing non-synonymous variants (Supplementary Data 5). For each GEA, we retrieved significantly associated loci by clustering significant variants within a distance of 10 kb. The significant variants clustered into 5–27 distinct loci per GEA consistent with a polygenic basis for most climate adaptation (Fig. S14–19; Supplementary Data 4). A polygenic architecture of thermotolerance was previously found in other fungi such as *Saccharomyces cerevisiae*[40,41]. A large number of associated loci were shared between GEA of different climatic variables. Highly correlated climatic variables including BIO5 and BIO10, the maximum temperature of the warmest month and the mean temperature of the warmest quarter, respectively, shared also higher numbers of significantly associated SNPs (Fig. 5d, Fig, S13a). However, we also identified hotspots of climatic adaptation loci for largely independent climatic variables (Fig. 5f). We identified a large segment of chromosome 7 and a telomeric region of chromosome 13 to be hotspots for climate associations. The chromosome 7 locus overlaps with the *Cyp51* gene involved in azole fungicide resistance. Hence, the association could be due to correlations in the application of fungicides and climatic factors. Temperate regions such as Europe show higher azole resistance (see above) than the Middle East, thus leading to an indirect association between fungicide resistance genes and climatic variables. However, at the same location on chromosome 7 is a quantitative trait locus (QTL) for growth at suboptimal temperature[42], so that locus could well underpin climatic adaptation independent of *Cyp51*. Further analyses of temperature sensitivity QTLs showed that three out of the four previously described loci are overlapping with loci associated with climatic variables (Supplementary Data 6). For example, the temperature-sensitivity QTL on chromosome 1 overlaps with loci associated with multiple temperature-related climatic variables such as the mean temperature of the warmest quarter (Fig. 6a–c). A variant associated with the mean temperature of the warmest quarter and overlapping with a QTL previously discovered in a central European cross (chr 1 at 2,090,068 bp) shows a global distribution of both alleles. By contrast, a second variant (chr 10 at 452,864 bp) associated with the mean temperature of the warmest quarter but not overlapping the previously discovered QTL (Fig. 6b, d), shows no allelic variation within central European populations. Alleles associated with warmer climates were also found in the US Midwest characterized by cold winters and high temperatures in summer. Our association analyses capture most likely only a subset of all loci contributing to climatic adaptation across the global distribution range. Hence, some degree of climate-genotype mismatches is expected across geographic regions if climatic adaptation is polygenic. Our analyses highlight the power of covering global genetic diversity to gain insights into the genetic architecture of recent adaptation in species.

To identify whether adaptive mutations tend to arise locally or occur across large geographic areas, we clustered significant loci based on the presence or absence of adaptive variants per country. We identified eight clusters characterized by shared adaptive variants with a similar geographic distribution (Fig. 6e, Fig. S19). Some clusters of adaptive variants are highly geographically localized (i.e., clusters 3, 5, and 6) while other clusters are widespread (i.e., cluster 4). Most adaptive alleles for extreme cold conditions were found in the populations subjected to the harsh winters of continental North America

(see cluster 3; Fig. 6e–g). Clusters of adaptive alleles were geographically widespread such as the distributions along the Mediterranean coast (see cluster 1). Taken together, the global genome panel revealed substantial standing variation for environmental adaptation with complex geographic patterns of local adaptation evolution.

## Discussion

Analysis of a thousand-genome panel recapitulated the spread of a major fungal crop pathogen revealing tight links to the history of global wheat cultivation. The early divergence between Middle Eastern and African genotypes from those collected in the rest of the world is consistent with a single expansion event from the center of origin dating back millennia. The extant genetic variation was strongly shaped by successive colonization bottlenecks during the introduction of the pathogen to the Americas and Oceania. The distinct loss of genetic diversity and increased linkage disequilibrium likely caused the loss of adaptive genetic variation and reduced evolutionary potential in the most recently colonized regions. TE activity has underpinned the rise of major adaptive mutations in the species[20,30,43]. Remarkably, the TE activity is underpinned by a marked relaxation or even loss of genomic defenses following population bottlenecks during global colonization. The higher activity of TEs is likely a direct consequence of reduced control and may have long-term consequences for the pathogen. The relaxation of genomic defenses in populations from the American, Oceanian, and European continents could have been selected as an evolvability trait, thus increasing variance in fitness in populations. The relaxation of genomic defenses likely underpins incipient genome expansion within the species while increasing the risk of mutational meltdown. The resilience of crops and agricultural ecosystems is threatened by the changing climate. The ability of pathogens to adapt and expand their range under altered humidity and temperature regimes as well as changes in seasonal patterns is a major concern. The identified genomic regions associated with adaptation to environmental conditions highlight how a global pathogen carries extensive variation to cope with climate change. Integrating population genetic information of pathogen adaptation to climatic gradients is a powerful asset for risk models of future pathogen spread. As climate change impacts crop production and most likely resistance to pathogens, future research explicitly addressing joint host and pathogen responses to warmer and drier climates is necessary to mitigate future crop losses.

## Methods

### Sample collection, culturing, and sequencing

Fungal isolates were grown on V8, yeast sucrose broth (YSB) or yeast malt agar (YMA) plates, or liquid culture prior to DNA extraction. Complete information about the geographic origin, date of collection, available sequencing datasets, and references are given in Supplementary Data 1. Lyophilized or frozen fungal tissue was used for DNA extraction with QIAGEN kits (DNeasy Plant Mini Kit, QIAcube HT). Sequencing libraries were prepared from sheared DNA on an Illumina sequencing platform following TruSeq library preparations. For the collection from the Joint Genome Institute (see Supplementary Data 1), DNA was extracted from single-spore isolates, the fragments were treated with end-repair, A-tailing, and ligation of Illumina-compatible adapters (IDT, Inc) using the KAPA-Illumina library creation kit (KAPA biosystems). Plate-based DNA library preparation for Illumina sequencing was performed on the PerkinElmer Sciclone NGS robotic liquid handling system using Kapa Biosystems library preparation kit. 200 ng of sample DNA was sheared to 600 bp using a Covaris LE220 focused-ultrasonicator. The sheared DNA fragments were size selected by double-SPRI and then the selected fragments were end-repaired, A-tailed, and ligated with Illumina-compatible sequencing adaptors from IDT containing a unique molecular index barcode for each sample

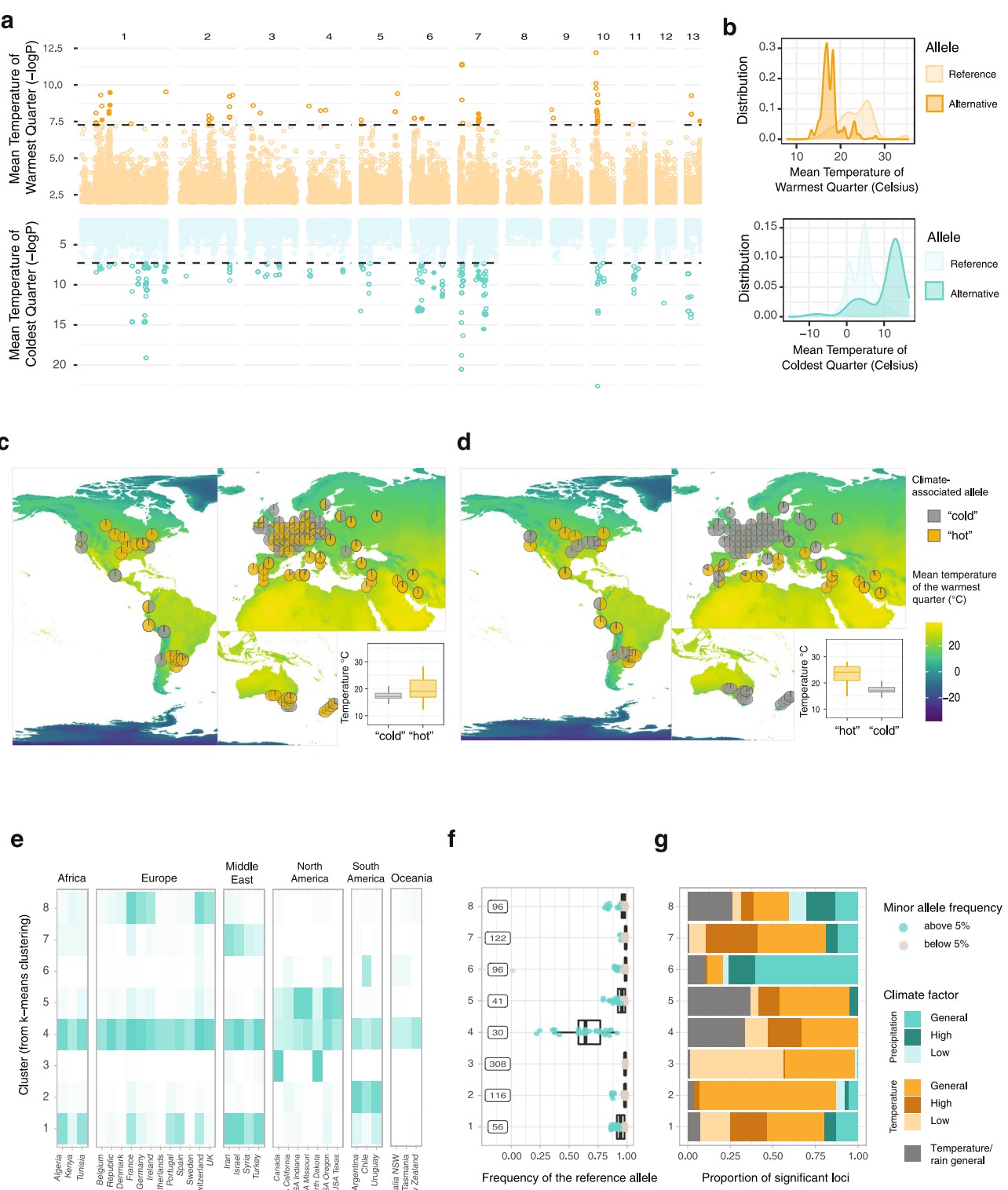

library. The prepared libraries were quantified using KAPA Biosystems' next-generation sequencing library qPCR kit and run on a Roche LightCycler 480 real-time PCR instrument. The quantified libraries were then prepared for sequencing on the Illumina HiSeq sequencing platform utilizing a TruSeq paired-end cluster kit, v4. Sequencing of the flow cell was performed on the Illumina HiSeq2500 sequencer using HiSeq TruSeq SBS sequencing kits, v4, following a 2 × 150 indexed run recipe. Overall, we obtained 1368 Illumina resequencing datasets for quality evaluation.

**Draft de novo assembly and variant calling procedures**

We used two different methods to investigate genetic variation. Firstly, we assembled the short-read sequencing datasets de novo using the software SPADES v.3.14.1[44] with the "careful" method (details of the assemblies in Supplementary Data 7). We used only the assemblies with fewer than 1600 contigs and a total assembly length between 30–40 Mb (filtered to remove any contigs shorter than 1 kb). Secondly, we called sequence variants using short-read mapping on the IPO323 reference genome. We trimmed and filtered the reads with

**Fig. 6 | Pathogen adaptation along global climatic gradients. a** Manhattan plot for genotype-climate associations for two bioclimatic variables related to high and low temperatures, respectively. Values obtained from the software GEMMA with the default linear model (Wald test). The horizontal line indicates the Bonferroni threshold. **b** Density plots for two examples of significantly associated variants from panel A. The lightest-colored curve represents the climatic values of the sampling site for the isolates carrying the reference allele and the darkest color for the isolates carrying the alternative allele. The variant shown in yellow is located on chromosome 10 at position 452,864 bp. The variant in teal is located on chromosome 6 at position 1,686,518 bp. **c** Map showing allele frequencies of a variant significantly associated with the mean temperature of the warmest quarter (chromosome 1, 2,090,068 bp, GEA based on 1103 isolates). The associated box plots represent the distribution of the mean temperature of the warmest cluster at the sampling location of isolate carrying the two alleles. The lower and upper hinges correspond to the first and third quartiles, the whiskers to the largest value

are within 1.5 times the inter-quartile range and the central horizontal line defines the median. **d** Identical to panel C for the variant at position 452864 bp on chromosome 10. **e** Heat map representing the proportion of minor alleles for all the variants in each k-means cluster which are present in each country (or state). Dark colored cells indicate that the minor allele is found at least once in the corresponding country (or state) for all the variants grouped in the corresponding k-means cluster. A white colored cell indicates that the minor allele is absent for all the variants classified in the k-mean cluster. **f** Minor allele frequency classified in each k-mean cluster. The framed numbers indicate the number of distinct variants included in each cluster. The lower and upper hinges of the boxplots correspond to the first and third quartiles, the whiskers to the largest value are within 1.5 times the inter-quartile range and the central horizontal line defines the median.
**g** Bioclimatic variables associated with the variants classified in each k-mean cluster.

Trimmomatic v.0.39, thereby removing adapter sequences, trimming leading and trailing bases with a quality lower than 15 for the resequencing of 2020 and 10 for all previous resequencing, and removing sequences shorter than 50 bp[45]. The trimmed reads were mapped to the reference genome of IPO323[24] using bowtie2 v.2.4.1[46]. GATK v4.1.4.1 was used for short-variant calling with the commands HaplotypeCaller, CombineGVCFs, and GenotypeGVCFs, setting the ploidy to 1 and the maximum number of alternative alleles to 2[47]. To filter out erroneously called short indels and SNPs, we started with a standard set of hard filters using the GATK quality metrics, for which the thresholds were set based on visualization of the metrics across the called variants. The per-site filters included: $FS > 10$, $MQ < 20$, $QD < 20$, ReadPosRankSum between $-2$ and 2, MQRankSum between $-2$ and 2, and BaseQRankSum between $-2$ and 2. We also included a per-genotype filter, removing any genotype with a depth lower than 3.

We further assessed the quality of our SNP calling using 8 isolates that were sequenced two times (including in some cases in different sequencing datasets). We assumed that the real variation between these pairs should be close to 0, although we cannot completely exclude the possibility of a small number of mutations happening during culturing or maintaining of these isolates in collection. We used the differences, i.e., erroneously called variants, between the resequencing pairs as an estimation of genotyping errors and to identify the causes of genotyping errors. Most of the erroneously called variants remaining after the hard filtering were related to genotypes with near-equal numbers of reads supporting the reference allele and an alternative allele, i.e., "heterozygous" alleles. Such genotypes were called with high confidence even though such a heterozygous-like pattern should be recognized as errors in a haploid organism and could be due to misalignment or repeated sequences in the genomes. We consequently implemented an allelic balance custom script to recognize such positions and to filter out any genotype that had fewer than 90% of reads supporting the called allele. This filter removed 75% of the erroneous variants left between the resequenced pairs after the hard filtering. As the rest of the erroneous variants were related to low sequencing depth, we further implemented a per-sample missing data and low-depth filtering, removing any sample with more than 20% of missing data and a mean depth of coverage lower than 6 on the core chromosomes (based on vcftools –missing-indv and –depth options)[48]. In the next filtering step, we removed the samples that were clones or near-clones. To identify these isolates, we created a network of isolates with an identity-by-state value superior to 0.99 (as measured by plink v1.9[49]) and extracted the subgraphs designating groups of clones (with the R packages tidygraph and ggraphs for visualization). In each group of clones, we filtered out all isolates except for the isolate with the lowest amount of missing data. These per-sample filters resulted in a final isolate count of 1109.

The final filtering step was a per-site filter based on the number of missing genotypes. Considering that *Z. tritici* contains accessory

chromosomes that are expected to be present in some isolates and absent in others, the relevant threshold of missing data had to be adapted per chromosome. To identify the presence-absence of accessory chromosomes, we assessed the depth of coverage in windows across all chromosomes with bedtools v.2.29.2 (option coverage followed by the option groupby to calculate the median per window)[50]. We normalized the depth estimates using the median depth over all windows of the core chromosomes for the per-window depth. The normalized depth was then used to infer presence-absence variants or copy-number variation of chromosomes, in which we considered that any chromosome with a normalized depth lower than 0.2 was absent. Based on the estimated number of chromosomes present in the dataset, we calculated missing data thresholds at 80% of genotyped isolates with the–max-missing-count option of vcftools ($NA_{max} = 222$ for the core chromosomes and between 328 and 1048 for the accessory chromosomes)[48].

## Population structure and population-level statistics
We used a subset of the filtered biallelic SNPs (one SNP every 1 kb, no missing data, and a minor allele frequency of 0.05) to estimate the population structure of our worldwide *Z. tritici* collection. This was done separately with a principal component analysis (R package SNPRelate[51]) and with a snmf clustering method from the LEA package[52]. The clustering analysis ran for a value of K (i.e., the number of clusters) ranging from 1 to 15 and with 10 repetitions per K. To identify the best K, we used the entropy method implemented in snmf which evaluates the quality of fit of the model to the data, as well as the smallest cluster size and the number of isolates assigned to any cluster with a coefficient higher than 0.75 (considered as non-admixed). We created a phylogenetic network with a subset of the isolates (7 randomly drawn isolates per country/state for all countries with at least 7 isolates), with the software SplitsTree5[53]. As an outgroup, we also included resequenced *Z. ardabiliae* isolates ("SRR6671804"-"SRR6671820")[54], for which the variant calling was performed using the same parameters and filters as above but only hard filters were applied (retained 11 isolates after filtering). For the network construction, SNPs were filtered to include only variants with no missing data in either *Z. ardabiliae* or *Z. tritici*, a MAF of 20%, and no variants closer than 1000 bp to reduce biases. We retained 9556 SNPs to compute the SplitsTree network.

For the analyses relying on the comparison of distinct groups, we discretized the populations by using only isolates belonging to one of the populations with a proportion higher than 0.75 at $K = 11$, the inferred best number of clusters. These genotypes were then used as input for treemix which infers splits between populations and creates a population tree[55]. We ensured that the tree shape was consistent regardless of possible migration events by running treemix with several possible migration events ranging from 0-6. We used the assembled genomes of *Z. ardabiliae* and *Z. passerinii* as outgroups to root the

tree[56]. We used the scikit-allel python package to measure genetic diversity (pi), taking into account only the non-admixed isolates from each cluster, in non-overlapping windows of 1 kb[57]. To remove windows with too much missing data (i.e., those that would artifactually lower the diversity), we selected only windows in which less than 20% of the variants were filtered out. We controlled for the variation in isolate numbers between clusters by subsampling each cluster 10 times to the smallest cluster size ($N = 16$) and averaging the obtained diversity estimates per window over the 10 subsamples.

### Transposable elements and repeat-induced point mutations

We called the TE insertions with the software ngs-te-mapper2[58]. In this process, the sequencing reads are first queried against a library of TE sequences, for which we used the TE consensus sequences obtained from 20 fully assembled genomes of 19 global isolates[59]. The "junction reads" that align both on a TE consensus sequence and on the flanking genome are used to determine the site of insertion of reference and non-reference TEs. To take any inaccuracies in the detection of the insertion sites into account, the positions of the insertions were rounded to 100 bp, so that insertions of the same element in a short window were considered to be the same insertion for further analyses. The insertions found in more than 10 samples were used to create a PCA (prcomp function from the stats R package), clustering the isolates based on their shared TE insertions.

We investigated the genomic distribution of variants, in relation to genomics, transcriptomics and epigenomics estimates. We gathered information from several sources and aggregated the data in 10-kb windows. We used transcriptomic data produced previously[60] and analyzed to calculate TPM[56], representing the gene expression during the necrotrophic and the asymptomatic phases of infection for the reference isolate. To include epigenomics data, we used previously published histone mark ChIP-seq data[26] and identified the peaks for several histone marks: H3K4me2, H3K27me3, and H3K9me2 (NCBI BioProject "PRJNA286790") assessed in two biological replicates. We trimmed and mapped the reads using Trimmomatic as above and bowtie, and filtered the mapped reads to keep only reads aligning with a quality higher than 30 using samtools. The histone mark peaks were called using macs2 (option −no-model)[61]. Only peak regions consistently identified in both replicates were kept (bedtools intersect). The coverage of histone mark peaks along chromosomes was computed by assessing the number of bases belonging to a histone mark peak per window. To remove windows with low variant counts due to a low mappability, we used genmap to estimate mappability and removed windows with a value lower than 0.85 (threshold estimated visually based on a density plot of windows across the genome)[62].

To analyze the repeat-induced point mutations in the collection of genomes, we used the same consensus TE sequences[59] as a reference genome for mapping of reads (as single reads) with bowtie2. Using a custom python script based on the biopython library[63], we also estimated the GC content and the RIP composite index[64], an estimate created to detect ratios of dimers indicative of mutations typical of the repeat-induced point mutation process (RIP). This was done on the reads aligning to the TE consensus sequences, thus providing an estimation of RIP in all transposable elements as well as per TE consensus. We also used previously computed values of the RIP composite index for 20 chromosome-level assemblies[29].

One of the most important enzymes for RIP is dim2. Based on previous knowledge that the strain Zt10 contains a functional copy of dim2[33], we extracted the sequence of the gene (Zt10_unitig_006_0417) as well as its two flanking genes (Zt10_unitig_006_0416, and Zt10_unitig_006_0418). These were then used as query sequences for the software *blast* to detect the presence and location of the 3 genes in de novo draft assemblies based on the Illumina resequencing. In many isolates dim2 is found in multiple copies. We thus identified the native copy as the copy found between the two flanking genes or, when the

assemblies did not include all three genes on one contig, the copy found within less than 10 kb of one of the flanking genes. We then considered the percentage of identity between the native copies and the functional copy of Zt10.

Statistical differences between geographical groups were assessed using a one-way ANOVA with blocks, with the sequencing batch considered as the confounding block. The sequencing batch effect was especially strong for the genomes from Hartmann et al.[65] probably due to a strong GC bias in the sequencing. As a post-hoc analysis, we used mean separation tests (least-square means) and displayed the results as letters on the corresponding plots.

### Adaptation and selection

We obtained geographical coordinates from the metadata attached to the isolates or inferred them based on the most precise sampling location available. Coordinates inferred can be found in Supplementary Data 1. We downloaded gridded weather and climate data at the 10' resolution from the WorldClim database version 2 (https://worldclim.org). The geographical coordinates were used to approximate the environmental conditions of origin for each isolate from all bioclimatic variables, which include for example the mean diurnal range and the annual precipitation as an average between 1970 and 2000. Based on these environmental estimates and the genomic variants, we identified genotype-to-environment associations with the software GEMMA 0.98.3[66]. We used a LOCO (Leave-One-Chromosome-Out) approach to estimate the kinship matrix on the genome excluding the chromosome on which we were estimating the associations. Significance threshold was set using the Bonferroni correction method, i.e., by dividing the traditional threshold of significance of 0.05 by the numbers of variants that were tested. Nearby significant SNPs were grouped together in "significant loci" if they were closer than 10 kb. To identify the genes which are potentially causal for adaptation to climatic conditions, we predicted the effect of the significant variants on the genes and proteins using SnpEff[67]. We created a custom SnpEff database so that the predictions would match the gene annotation we are using[68] and setting the upstream/downstream interval length to 1 kb. We also used the SnpEff predictions to compare the distribution of effects (synonymous, non-synonymous, and modifier) in the significantly associated variants and in all the variants using 200 random draws.

We investigated the geographic distribution of the potentially adaptive alleles we found through k-means clustering of each allele's presence-absence per country. Per significant locus and per bioclimatic variable, we selected the variant with the lowest $p$-value (i.e., the top variant in each "peak"), and identified whether the minor allele was present or absent per country/state including more than 5 isolates. The matrix of presence/absence was then used for the clustering. This analysis did not reveal a clear-cut pattern of adaptation sharing a set number of geographic distributions. Although there was no clear-cut best number of clusters even when testing up to 30 clusters, we chose eight clusters for graphical representation, using the elbow method. To investigate fungicide resistance, we identified the presence/absence of known resistance alleles in the isolates of the dataset, following the method in ref. [15]. We then compared the frequency of these alleles in the different geographic locations, and through time.

### Reporting summary

Further information on research design is available in the Nature Portfolio Reporting Summary linked to this article.

## Data availability

All sequencing data is available from the NCBI Sequence Read Archive. Individual accession numbers can be retrieved from Supplementary Data 1 and from the Methods section. Climatic data was obtained from the publicly available WorldClim database version 2.

## Code availability

To ensure reproducibility of the analyses presented in this manuscript, all custom scripts are available from https://github.com/afeurtey/WW_PopGen (https://doi.org/10.5281/zenodo.7572234)[69]. Post-processing and visualization of the data were done in R, bash, and python, available as R markdown reports in the github repository.

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

## Acknowledgements

Data produced and analyzed in this paper were generated in collaboration with the Genetic Diversity Centre (GDC), ETH Zurich. We would like to thank Lucio Garcia and the NGS platform at Syngenta for assistance with sequencing. DC and GS were supported by the Swiss Innovation Agency Innosuisse. The work (proposal: 10.46936/10.25585/60000699) conducted by the U.S. Department of Energy Joint Genome Institute (https://ror.org/04xm1d337), a DOE Office of Science User Facility, is supported by the Office of Science of the U.S. Department of Energy operated under Contract No. DE-AC02-05CH11231 and lab work was supported by the USDA-Agricultural Research Service project 3602-22000-015-00D. AF was supported by a grant from the DFG priority programme SPP1819 awarded to EHS.

## Author contributions

A.F. and D.C. conceived the study; A.F. and C.L. performed analyses; A.F., M.C.M., A.M., P.S.S., R.W., G.P., G.S., S.F.F.T., L.G., T.C.M., F.S., J.A., A.L., Y.Y., C.D., K.B., I.V.G., S.B.G., A.G., M.F.S., E.H.S., M.H.L., M.H.L., G.H.J.K., B.A.M., and D.C. contributed samples and/or genome sequencing data; B.A.M. and D.C. supervised the work; A.F. and D.C. wrote the manuscript with input from all co-authors.

## Competing interests

The authors declare no competing interests.
