## [Peer Review File · Nature Communications]

A thousand-genome panel retraces the global spread and adaptation of a major fungal crop pathogenREVIEWER COMMENTS

Reviewer #1 (Remarks to the Author):

- The study examines a pathogen's diversification as its host was domesticated around the globe. Cultures of >1100 strains of the wheat pathogen *Z. tritici* were sequenced and subjected to genome-wide genotyping using SNPs. Major findings of this global analysis was the huge proliferation of transposable elements in the genomes of pathogens isolated from the Americas and Oceania, which seem to have been released from the efficacy of RIP seen in the European and native ranges of the host. The authors also used genotype by environment analysis to identify variants that were associated with certain climatic variables include high and low mean temperature among others.
- No study has examined whole-genome population genetics in a plant pathogenic fungus on this scale. We know so little about global-scale fungal population genomics, this study will add a great deal to our field.
- Overall, this study is clear, well written, and their findings are well presented. My comments and edits (below) are minor. I feel this is a very high quality manuscript. If anything, the study is almost two separate studies (one examining the genome evolution of this pathogen of a domesticated host, and another examining genome adaptation along climatic and antifungal variables).
- I think the methods are sound and their conclusions are supported.
- The methods are thoroughly documented, the author's github website provided has the information and files already available. Raw data should be available to the public through the SRA.

Specific comments:

Figure 1A - many of the pie charts here are really small. Could this not be it's own figure for the sake of clarity?

Line 186: "Due to the deleterious nature of TEs" should have a reference.

Line 230: I think you need to add a clause to this sentence to clarify why you performed this analysis, accounting for sequencing depth. It will add clarity for those not used to thinking about the intersection of short read sequencing assembly and TEs.

Line 240: "evidence of RIP-like mutations by reporting the..."

Figure 3F & Lines 246-248: What about the European data? It's not mentioned here that this signature is missing in the EU samples, where there is a huge amount of variation in the number of TE insertions with a pretty conserved amount of RIP composite index median.

Figure 4A: Text too small. In general text in figures 4 and 5 could be made much larger.

Figure 5E: No scale for heatmap.

Lines 345 and 348: "nonsynonymous" and "non-synonymous" pick one.

Line 350: "consistent with a polygenic basis for most climate adaptation." Have studies of other fungi found this? If so, cite here.

Figure 5D and Lines 369 - 373: I find it odd that the "hot" allele is most common/exclusively found in upper mid North America (Canada, North Dakota).

Lines 401 - 403: It may be worth mentioning here that pathogen fitness/ adaptability will intersect with the host's response to a changing climate as well, which could have significant impact on crop production and the genetic variation in the host. So for example, less hot-adapted plant genotypes might be more susceptible to pathogen attack, which could accelerate the removal of non hot-adapted genotypes from the wheat population.

Reviewer #2 (Remarks to the Author):

A hugely impressive paper, with many suitable and informative analyses and insights into the global emergence and evolution of the important plant pathogen *Z. tritici*.

Below are my recommendations for improvement and/or clarification, in no particular order:

In the introduction, you write "weakened genomic defences" and later on line 186 "genomic defences have evolved an array of mechanisms to counteract their activity". Can you be clear about the mechanisms you are alluding to? Just RIP or others?

line 100: "We gathered 1109 high-quality, whole-genome, short-read sequencing datasets.". You mean assembled. also, they can no longer be considered "high quality" if they are generated from short-read sequencing datasets alone? On line 434 and later you write "We created de novo draft assemblies" - again that should be assembled- and it sounds like you should write 'draft' throughout based on your sequencing and assembly strategy. I also do not see details of each of these assemblies in terms of contig count, N50, total size etc. There should be a sup table of these details.

Provide further details in discussion on how the genetic exchange you have identified is likely occurring.

Fig S3. Have you not also made a cross-validation error plot to show the most likely value for k. This would make a useful additional sup plot.

Importantly, I would like to see a standard phylogenetic tree made using RAxML, NJ or other of all the isolates in the supplemental material. This could either be midpoint rooted, or if possible, rooted to an outgroup relative. Or both would be most informative.

Fig S7. missing the 'B' label. Can you also define further how a TIP was found to be shared between isolates? did it need the exact coordinates, or was any overlap allowed? For the text, how many were identified in each isolate across all populations (i.e., range, mean and median counts?). Did any populations have more than others? (This seems to be described in lines 225-234 but not with a table of counts or any specific counts mentioned)

200: For H3K9me2 and me3 marks, you should include project accession numbers for the data used. Also, it is unclear how you define hetero and euchromatin regions based on those peaks? how far from a peak does a region become classified? And what about the control/input data? was that used in peak calling?

Another further analysis I think is essential, is to define the family or groups of TEs you are identifying? there are many types, and this should be identified/characterized, perhaps using a program like RepeatModeller/RepeatMasker. Important questions include if they are all the same type across all populations? What the baseline repetitive content of the core and accessory chromosomes? Of those new TE's, do they have all domains required to be active? (i.e., determine which are active and which are not). Also, how are the 'new' ones distinguished from the 'old ones'... Are you just reporting on anything deviating from a single reference strain?

Reviewer #3 (Remarks to the Author):

The authors have submitted an investigation of global diversity and patterns of biogeography and local adaptation in the wheat pathogen *Z. tritici*. With over 1000 genomes, across the entire global range, the study is robust. The observation of importance of RIP and TEs is interesting and suggests a mechanism of genome evolution that should be investigated in other pathogens, in particular those with recent expansion into new geographic regions.

The paper is well written, and methods support conclusions.

A few minor edits are needed (odd wording and errors in figure legends primarily) but overall I see nothing major that needs to be addressed.

Minor comments:

1. what is the overall recombination rate in this pathogen? How often is sex occurring? If this is known/estimated I think this is an important fact that should be added
2. are there any major lineages that might be missed in the sampling and why (geopolitical or other issues)? do the authors think this might impact results? I'm thinking specifically of large regions of wheat cultivation in Ukraine, Russia and China.

We included a manuscript file version showing track changes in yellow as "Related Manuscript File".

#####

Response to Reviewers

Reviewer #1 (Remarks to the Author):

- *The study examines a pathogen's diversification as its host was domesticated around the globe. Cultures of >1100 strains of the wheat pathogen *Z. tritici* were sequenced and subjected to genome-wide genotyping using SNPs. Major findings of this global analysis was the huge proliferation of transposable elements in the genomes of pathogens isolated from the Americas and Oceania, which seem to have been released from the efficacy of RIP seen in the European and native ranges of the host. The authors also used genotype by environment analysis to identify variants that were associated with certain climatic variables include high and low mean temperature among others.*
- *No study has examined whole-genome population genetics in a plant pathogenic fungus on this scale. We know so little about global-scale fungal population genomics, this study will add a great deal to our field.*
- *Overall, this study is clear, well written, and their findings are well presented. My comments and edits (below) are minor. I feel this is a very high quality manuscript. If anything, the study is almost two separate studies (one examining the genome evolution of this pathogen of a domesticated host, and another examining genome adaptation along climatic and antifungal variables).*
- *I think the methods are sound and their conclusions are supported.*
- *The methods are thoroughly documented, the author's github website provided has the information and files already available. Raw data should be available to the public through the SRA.*

Specific comments:

Figure 1A - many of the pie charts here are really small. Could this not be it's own figure for the sake of clarity?

Thank you for these comments. We have reconsidered all our display elements and have divided some to create more space. For Figure 1, we prefer to keep this figure as one unit so that the panels can be easily compared. The most important information in this panel is represented by colors and all the details can be found in the supplementary tables. We feel that the main message is still best conveyed by this composite layout.

Line 186: "Due to the deleterious nature of TEs" should have a reference.

We have added the corresponding citation (both for the first part of this sentence and the second). We also adjusted the sentence according to a comment made by Reviewer 2.

Line 230: I think you need to add a clause to this sentence to clarify why you performed this analysis, accounting for sequencing depth. It will add clarity for those not used to thinking about the intersection of short read sequencing assembly and TEs.

We clarified the sentence accordingly.

Line 240: "evidence of RIP-like mutations by reporting the..."

Corrected

Figure 3F & Lines 246-248: What about the European data? It's not mentioned here that this signature is missing in the EU samples, where there is a huge amount of variation in the number of TE insertions with a pretty conserved amount of RIP composite index median.

The range of the RIP composite index is indeed small within the European population. However, the correlation between number of TEs and RIP is significant among European samples although the slope is flatter. We have added these clarifications to the text, as suggested.

Figure 4A: Text too small. In general text in figures 4 and 5 could be made much larger.

We have increased the font size by 1-2 points wherever possible in these two figures.

Figure 5E: No scale for heatmap.

The heatmap scale is meant in a qualitative and we clarified the interpretation of the color scale in the legend.

Lines 345 and 348: "nonsynonymous" and "non-synonymous" pick one.

Corrected.

Line 350: "consistent with a polygenic basis for most climate adaptation." Have studies of other fungi found this? If so, cite here.

We are not aware of comparable studies on the genetic basis of climatic adaptation in filamentous fungi but we cite now studies related to thermotolerance in yeasts. A polygenic landscape of adaptation to climate-related factors such as aridity was found in plants (e.g. <https://doi.org/10.1111/mec.14723>). As these studies are not directly linked to our findings, we have chosen to not cite these.

Figure 5D and Lines 369 - 373: I find it odd that the "hot" allele is most common/exclusively found in upper mid North America (Canada, North Dakota).

This is an excellent comment and we agree that this seems counterintuitive. However, some parts of central North America have a continental climate characterized by a very high seasonal variance with very cold winter but also high temperatures in summer. It is expected to find some alleles related to adaptation to high temperature in these regions, especially when investigating values about the warmest times of the year such as in figure 5D, rather than annual averages. In addition, we are most likely capturing only a portion of the loci contributing to climatic adaptation. Hence, some degree of climate-genotype mismatches is expected across geographic regions if climatic adaptation is polygenic. We more specifically address now the apparent mismatch in the relevant results section.

Lines 401 - 403: It may be worth mentioning here that pathogen fitness/ adaptability will intersect with the host's response to a changing climate as well, which could have significant impact on crop production and the genetic variation in the host. So for example, less hot-adapted plant genotypes might be more

susceptible to pathogen attack, which could accelerate the removal of non hot-adapted genotypes from the wheat population.

Yes, we completely agree. We have added a sentence acknowledging the importance of the host response to climate change.

Reviewer #2 (Remarks to the Author):

*A hugely impressive paper, with many suitable and informative analyses and insights into the global emergence and evolution of the important plant pathogen *Z. tritici*.*

Thank you. We are very grateful for the insightful suggestions below.

Below are my recommendations for improvement and/or clarification, in no particular order:

In the introduction, you write "weakened genomic defences" and later on line 186 "genomic defences have evolved an array of mechanisms to counteract their activity". Can you be clear about the mechanisms you are alluding to? Just RIP or others?

We have added more details in the results section including relevant citations.

line 100: "We gathered 1109 high-quality, whole-genome, short-read sequencing datasets.". You mean assembled. also, they can no longer be considered "high quality" if they are generated from short-read sequencing datasets alone? On line 434 and later you write "We created de novo draft assemblies" - again that should be assembled- and it sounds like you should write 'draft' throughout based on your sequencing and assembly strategy. I also do not see details of each of these assemblies in terms of contig count, N50, total size etc. There should be a sup table of these details.

We used several different strategies to generate different elements of the genetic variation data. In the population structure section, we have used solely variant calling by read mapping and not assembly-based approaches. This was done on 1300+ samples, but only 1109 passed our quality filtering described in the methods, hence the term "high quality" sequencing sets. We also separately assembled the Illumina sequencing datasets *de novo* to produce draft assemblies. This data that was used for example to extract the *dim2* sequences from all isolates. We have clarified this in the methods section and in the text at the relevant line as suggested. We also have added a supplementary table S7 providing information about the draft assemblies.

Provide further details in discussion on how the genetic exchange you have identified is likely occurring.

We now mention that recent and historic trade is most likely responsible for patterns of genetic exchange.

Fig S3. Have you not also made a cross-validation error plot to show the most likely value for k . This would make a useful additional sup plot.

Yes, we have indeed. The cross-entropy plots and other information concerning the choice of the most likely value for k are supplied in Fig. S2.

Importantly, I would like to see a standard phylogenetic tree made using RAxML, NJ or other of all the isolates in the supplemental material. This could either be midpoint rooted, or if possible, rooted to an outgroup relative. Or both would be most informative.

Yes, alternative representations of the variant call data are indeed useful. As *Z. tritici* undergoes frequent sexual reproduction and has a high recombination rate, we chose to generate a network tree (SplitsTree) to represent branch lengths. A network captures intra-specific relationships much more accurately than a standard phylogenetic tree. Recombination among haplotypes would have significantly inflated branch lengths in a tree leading to erroneous interpretations of genetic distances. We have added a supplementary figure S4 including *Z. ardabiliae* sister species isolates as outgroup as suggested.

Fig S7. missing the 'B' label. Can you also define further how a TIP was found to be shared between isolates? did it need the exact coordinates, or was any overlap allowed?

Yes, we allowed for some overlap to avoid erroneously splitting TIP loci due to minor imprecisions in the localizations. We have added this information to the results text also to make this point clearer.

For the text, how many were identified in each isolate across all populations (i.e., range, mean and median counts?). Did any populations have more than others? (This seems to be described in lines 225-234 but not with a table of counts or any specific counts mentioned)

We have added the specific count ranges and medians in the main text, taking into account the sequencing depth.

200: For H3K9me2 and me3 marks, you should include project accession numbers for the data used. Also, it is unclear how you define hetero and euchromatin regions based on those peaks? how far from a peak does a region become classified? And what about the control/input data? was that used in peak calling?

We have added the Bioproject accession number for the ChIP-seq data in addition to the citation for the original source. We have not classified the windows in hetero/euchromatin categories, but instead have used peak calls (and their breadth) in windows along chromosomes. We have clarified how we have used the peak information in our methods text.

The original publication did not produce control or input data for the ChIP-seq experiment but rather provided two biological replicates for each mark. We analyzed each replicate and reported only regions with overlapping peak calls. We have clarified this in the methods.

Another further analyses I think is essential, is to define the family or groups of TEs you are identifying? there are many types, and this should be identified/characterized, perhaps using a program like RepeatModeller/RepeatMasker. Important questions include if they are all the same type across all populations?

Yes, we had previously performed detailed annotation and classification of the TEs of *Z. tritici* (see Badet et al. BMC Biology 2020). The TE library used in the current manuscript is based on a combined and curated set of TEs discovered across 19 completely assembled genomes of the same species.

We have added the details of the superfamilies and families that were most found to be specifically responsible for the expansion. We have added information about which families are shared between population and which elements show a population-specific expansion. These results are described in the main text as well as in additional supplementary figures.

What the baseline repetitive content of the core and accessory chromosomes?

We have added the % repeat content for core and accessory chromosomes in the results text.

Of those new TE's, do they have all domains required to be active? (i.e., determine which are active and which are not). Also, how are the 'new' ones distinguished from the 'old ones'... Are you just reporting on anything deviating from a single reference strain?

We are reporting insertions discovered in relationship to the reference genome using the widely used approach to analyze short read mapping patterns against the genome and a TE consensus sequence library. We were able to assign newly identified TEs to a specific TE already discovered in the species (see above), however we are unable to analyze the specific sequence of the inserted TE as the short read-based method is unable to assemble entire TE sequences given the repetitive nature. This also means that we are unable to confidently detect specific modifications in newly active TEs given the limitations of short read genomes. We do provide now a more detailed representation of what TE families (and superfamilies) contributed to the increase in TEs in the species. Furthermore, we have clarified the approach in the results.

Reviewer #3 (Remarks to the Author):

The authors have submitted an investigation of global diversity and patterns of biogeography and local adaptation in the wheat pathogen *Z. tritici*. With over 1000 genomes, across the entire global range, the study is robust. The observation of importance of RIP and TEs is interesting and suggests a mechanism of genome evolution that should be investigated in other pathogens, in particular those with recent expansion into new geographic regions.

The paper is well written, and methods support conclusions.

A few minor edits are needed (odd wording and errors in figure legends primarily) but overall I see nothing major that needs to be addressed.

Minor comments:

1. what is the overall recombination rate in this pathogen? How often is sex occurring? If this is known/estimated I think this is an important fact that should be added

Sexual reproduction has been estimated to be happening mostly at the end of the wheat growing season but also during the season itself (Zhan et al, 1998). Recombination rates are high in this pathogen as was estimated both computationally and experimentally (Croll et al. 2015, Stukenbrock et al. 2019). We have added this information in the text along with the network tree following a suggestion from reviewer 2.

2. are there any major lineages that might be missed in the sampling and why (geopolitical or other issues)? do the authors think this might impact results? I'm thinking specifically of large regions of wheat cultivation in Ukraine, Russia and China.

Interestingly, Septoria blotch disease has not been widely reported in China despite, in principle, favorable conditions (see DOI: [10.19103/AS.2021.0092.10](https://doi.org/10.19103/AS.2021.0092.10)). It is true, however, that we have weak coverage in our sampling of Russia, the Balkans, and Ukraine. It would have been interesting to obtain a more detailed view of the contact zones between the European cluster and the ones from the Middle-East in terms of phylogeography and gene flow. Although we recognize that this is a weakness for the completeness of our study, we do not think that a more comprehensive sampling in these areas would have affected our conclusions. Hopefully, the sampling can be improved once the geopolitical situation in this area has improved. We mention now specific gaps in the sampling in the results text.

REVIEWERS' COMMENTS

Reviewer #2 (Remarks to the Author):

Thank you for addressing my comments made in the first round of revisions. I am satisfied by all the comments and revisions made.